# Bus Violence: An Open Benchmark for Video Violence Detection on Public Transport

**DOI:** 10.3390/s22218345

**Published:** 2022-10-31

**Authors:** Luca Ciampi, Paweł Foszner, Nicola Messina, Michał Staniszewski, Claudio Gennaro, Fabrizio Falchi, Gianluca Serao, Michał Cogiel, Dominik Golba, Agnieszka Szczęsna, Giuseppe Amato

**Affiliations:** 1Institute of Information Science and Technologies, National Research Council, Via G. Moruzzi 1, 56124 Pisa, Italy; 2Department of Computer Graphics, Vision and Digital Systems, Faculty of Automatic Control, Electronics and Computer Science, Silesian University of Technology, Akademicka 2A, 44-100 Gliwice, Poland; 3Department of Information Engineering, University of Pisa, Via Girolamo Caruso, 16, 56122 Pisa, Italy; 4Blees sp. z o.o., Zygmunta Starego 24a/10, 44-100 Gliwice, Poland

**Keywords:** violence detection, action recognition, fight detection, video surveillance, deep learning, violence detection benchmark, violence on public transport

## Abstract

The automatic detection of violent actions in public places through video analysis is difficult because the employed Artificial Intelligence-based techniques often suffer from generalization problems. Indeed, these algorithms hinge on large quantities of annotated data and usually experience a drastic drop in performance when used in scenarios never seen during the supervised learning phase. In this paper, we introduce and publicly release the *Bus Violence* benchmark, the first large-scale collection of video clips for violence detection on public transport, where some actors simulated violent actions inside a moving bus in changing conditions, such as the background or light. Moreover, we conduct a performance analysis of several state-of-the-art video violence detectors pre-trained with general violence detection databases on this newly established use case. The achieved moderate performances reveal the difficulties in generalizing from these popular methods, indicating the need to have this new collection of labeled data, beneficial for specializing them in this new scenario.

## 1. Introduction

The ubiquity of video surveillance cameras in modern cities and the significant growth of Artificial Intelligence (AI) provide new opportunities for developing functional smart Computer Vision-based applications and services for citizens, primarily based on deep learning solutions. Indeed, on the one hand, we are witnessing an increasing demand for video surveillance systems in public places to ensure security in different urban areas, such as streets, banks, or railway stations. On the other hand, it has become impossible or too expensive to manually monitor this massive amount of video data in real time: problems such as a lack of personnel and slow response arise, leading to the strong demand for automated systems.

In this context, many smart applications, ranging from crowd counting [1,2] and people tracking [3,4] to pedestrian detection [5,6], re-identification [7], or even facial reconstruction [8], have been proposed and are nowadays widely employed worldwide, helping to prevent many criminal activities by exploiting AI systems that automatically analyze this deluge of visual data, extracting relevant information. In this work, we focus on the specific task of violence detection in videos, a subset of human action recognition that aims to detect violent behaviors in video data. Although this task is crucial to investigate the harmful abnormal contents from the vast amounts of surveillance video data, it is relatively unexplored compared to common action recognition.

One of the potential places in which an automatic violence detection system should be developed is public transport, such as buses, trains, etc. However, evaluating the existing approaches (or creating new ones) in this scenario is difficult due to the lack of labeled data. Although some annotated datasets for video violence detection in general contexts already exist, the same cannot be said for the case of public transport environments. To fill this gap, in this work, we introduce a benchmark specifically designed for this scenario. We collected and publicly released [9] a large-scale dataset gathered from multiple cameras located inside a moving bus where several people simulated violent actions, such as stealing an object from another person, fighting between passengers, etc. Our dataset, named *Bus Violence*, contains 1400 video clips manually annotated as having (or not) violent scenes. To the best of our knowledge, it is the first dataset entirely located on public transport and is one of the biggest benchmarks for video violence detection in the literature. The main difference compared to the other existing databases is also connected to the dynamic background—the violent actions are recorded during bus movement, which indicates a different illumination (in contrast to the static background of other datasets), making violence detection much more challenging.

In this paper, we first introduce the dataset and describe the data collection and annotation processes. Then, we present an in-depth experimental analysis of the performance of several state-of-the-art video violence detectors in this newly established scenario, serving as baselines. Specifically, we employ our *Bus Violence* dataset as a testing ground for evaluating the generalization capabilities of some of the most popular deep learning-based architectures suitable for video violence detection, pre-trained over the general violence detection databases present in the literature. Indeed, the *Domain Shift* problem, i.e., the domain gap between the train and the test data distributions, is one of the most critical concerns affecting deep learning techniques, and it has become paramount to measure the performance of these algorithms against scenarios never seen during the supervised learning phase. We hope this benchmark and the obtained results may become a reference point for the scientific community concerning violence detection in videos captured from public transport.

Summarizing, the contributions of this paper are three-fold:We introduce and publicly release [9] the *Bus Violence* dataset, a new collection of data for video violence detection on public transport;We test the generalization capabilities over this newly established scenario by employing some state-of-the-art video violence detectors pre-trained over existing general-purpose violence detection data;We demonstrate through extensive experimentation that the probed architectures struggle to generalize to this very specific yet critical real-world scenario, suggesting that this new collection of labeled data could be beneficial to foster the research toward more generalizable deep learning methods able to also deal with very specific situations.

The rest of the paper is structured as follows. Section 2 reviews the related work on the existing datasets and methods for video violence detection. Section 3 describes the *Bus Violence* dataset. The performance analysis of several popular video violence detection techniques on this newly introduced benchmark is presented in Section 4. Finally, we conclude the paper with Section 5, suggesting some insights on future directions. The evaluation code and all other resources for reproducing the results are available at https://ciampluca.github.io/bus_violence_dataset/ (accessed on 20 September 2022).

## 2. Related Work

Several annotated datasets have been released in the last few years to support the supervised learning of modern video human action detectors based on deep neural networks. One of the biggest datasets was proposed in the project of *Kinetics 400/600/700* [10,11,12] related to the number of human action classes, such as people interactions and single behavior. The given benchmark consists of high-quality videos of about 650,000 clips lasting around 10 s each. Alternatively, other options are represented by *HMDB51* [13], which consists of nearly 7000 videos recorded for 51 action classes, and *UCF-101* [14], made up of 101 action classes over 13 k clips and 27 h of video data. In contrast, datasets containing only abnormal actions (such as fights, robberies, or shootings) were introduced in the *UCF-Crime* benchmark [15], a large-scale dataset of 1900 real-world surveillance videos for anomaly detection.

However, in the literature, there are only a few benchmarks suitable for the video violence detection task, which consists of binary classifying clips as containing (or not) any actions considered to be violent. In [16], the authors introduced two video benchmarks for violence detection, namely the *Hockey Fight* and the *Movies Fight* datasets. The former consists of 200 clips extracted from short movies, a number that is insufficient nowadays. On the other hand, the second one has 1000 fight and non-fight clips from the ice hockey game. In this case, the lack of diversity represents the main drawback because all the videos are captured in a single scene. Another dataset, named *Violent-Flows*, has been presented in [17]. It consists of about 250 video clips of violent/non-violent behaviors in general contexts. The main peculiarity of this data collection is represented by its overcrowded scenes but low image quality. Moreover, in [18], the *NTU CCTV-Fights* is introduced, which covers 1000 videos of real-world fights coming from CCTV or mobile cameras.

More recently, the authors of [19,20] proposed the *AIRTLab* dataset, a small collection of 350 video clips labeled as “non-violent” and “violent,” where the non-violent actions include behaviors such as hugs and claps that can cause false positives in the violence detection task. Furthermore, the *Surveillance Camera Fight* dataset has been presented in [21]. It consists of 300 videos in total, 150 of which describe fight sequences and 150 depict non-fight scenes, recorded from several surveillance cameras located in public spaces. Moreover, the *RWF-2000* [22] and the *Real-Life Violence Situations* [23] datasets consist of video gathered from public surveillance cameras. In both collections, the authors collected 2000 video clips: half of them include violent behaviors, while the others belong to non-violent activities. All these benchmarks share the characteristic of having a still background because the clips are captured from fixed surveillance cameras. We summarize the statistics of all the above-described databases in Table 1.

To complement these datasets, in this work, a new large-scale benchmark suitable for human violence detection is constructed by gathering video clips from several cameras located inside a moving bus. To the best of our knowledge, our *Bus Violence* dataset is the first collection of videos depicting violent scenes concerning public transport.

## 3. The Bus Violence Dataset

Our *Bus Violence* dataset [9] aims to overcome the lack of significant public datasets for human violence detection on public transport, such as buses or trains. Already published benchmarks mainly present situations with actions in stable conditions from videos gathered by urban surveillance cameras located in fixed positions, such as buildings, street lamps, etc. On the other hand, records on public transport change in many directions: (1) the background outside windows have a different view due to general movement, (2) the movement is dynamic, but it can be slow or fast, and (3) there are many illumination changes due to different weather conditions and the position of the vehicle. For those reasons, the proposed *Bus Violence* benchmark consists of data recorded in dynamic conditions (general bus movement). In the following, we detail the processes of the data collection and curation.

### 3.1. Data Collection

The videos were acquired in a three-hour window during the day, during which the bus continued traveling and stopping around closed zones. The participants of the records were getting inside and outside the bus, playing already defined actions. Specifically, the unwanted situations (treated as violent actions) were concerned as a fight between passengers, kicking and tearing pieces of equipment, and tearing out or stealing an object from another person (robbery). An important aspect is the diversity of people. Ten actors took part in the recordings and changed their clothes at different times to ensure a reliable variety of situations. In addition, thanks to the conditions in the closed depot, it was possible to obtain different lighting conditions, for example, driving in the sun, parking in a very shaded place, etc.

The test system was able to record videos from three cameras at 25 FPS in *.mp4* format (H.264). Our recording system was installed manually by us and composed of two cameras located in the corners of the bus (with resolution 960×540 and 352×288 px, respectively) and one fisheye in the middle (1280×960 px). In total, we recorded a three-hour video—one hour dedicated to actions considered violent and two hours to non-violent situations.

### 3.2. Data Curation

After the acquisition, collected videos were manually checked and split. Specifically, we divided all the videos into single shorter clips, ranging from 16 frames to a maximum length of 48 frames, capturing an exact action (either violent or non-violent). This served to avoid single shots containing both violent and non-violent actions, which may be confusing for video-level violence detection models. Then, these resulting videos were filtered and annotated. In particular, the ones not containing a violent action were classified as non-violent situations. In these clips, passengers were just sitting, standing, or walking inside a bus. More in-depth, we operated by exploiting a two-stage manual labeling procedure. In the first step, three human annotators performed a preliminary video classification into the two classes—violence/no violence. Then, in the second stage, two additional independent experts conducted further analysis, filtering out the wrong samples. To obtain more reliable labels, we decided not to leverage the use of automatic labeling tools that would have required further manual verification.

After the above-described operations, the non-violence class resulted in more videos than the violence class. Therefore, we undersampled the non-violence samples by randomly discarding videos to balance the dataset perfectly. In the end, the final curated dataset contains 1400 videos, evenly divided into the two classes. In each class, we obtained almost the same number of videos for each of the three different resolutions. Specifically, we obtained 212 violence and 240 non-violence clips for the 1280×960 px resolution, 222 violence and 210 non-violence for the 960×540 px resolution, and 266 violence and 250 non-violence for the 352×288 px resolution. We placed them in two separate folders, each containing 700 *.mp4* video files encoded in the H.264 format. We report the final statistics of the resulting dataset in Table 2.

In Figure 1 and Figure 2, we show some samples from the final curated dataset of the violence and non-violence classes, respectively.

## 4. Performance Analysis

In this section, we evaluate several deep learning-based video violence detectors present in the literature on our *Bus Violence* benchmark. Following the primary use case for this dataset explained in Section 1, we employ it as a test benchmark (although in this work we exploited the whole dataset as a test benchmark, in [9], we provide training and test splits for researchers interested in also using our data for training purposes) to understand how well the considered methods, pre-trained over existing general violence detection datasets, can generalize to this very specific yet challenging scenario.

### 4.1. Considered Methods

We selected some of the most popular methods coming from human action recognition, adapting them to our task, and some of the most representative techniques specific to video violence detection. We briefly summarize them below. We refer the reader to the papers describing the specific architectures for more details.

Human action recognition methods aim to classify videos in several classes, relying on the human actions that occur in them. Because actions can be formulated as spatiotemporal objects, many architectures that extend 2D image models to the spatiotemporal domain have been introduced in the literature. Here, we considered the ResNet 3D network [24] that handles both spatial and temporal dimensions using 3DConv layers [25] and the ResNet 2+1D architecture [24], which instead decomposes the convolutions into separate 2D spatial and 1D temporal filters [26]. Furthermore, we took into account SlowFast [27], a two-pathway model where the first one is designed to capture the semantic information that can be given by images or a few sparse frames operating at low frame rates, while the other one is responsible for capturing rapidly changing motion by operating at a fast refreshing speed. Finally, we exploited the Video Swim Transformer [28], a model that relies on the recently introduced Transformer attention modules in processing image feature maps. Specifically, it extends the efficient sliding-window Transformers proposed for image processing [29] to the temporal axis, obtaining a good efficiency-effective trade-off.

On the other hand, video violence detection methods aim at binary classifying videos to predict if they contain (or not) any actions considered to be violent. In this work, we exploited the architecture proposed in [30], consisting of a series of convolutional layers for spatial features extraction, followed by Convolutional Long Short Memory (ConvLSTM) [31] for encoding the frame-level changes. Furthermore, we also considered the network in [32], a variant of [30], where a spatiotemporal encoder built on a standard convolutional backbone for features extraction is combined with the Bidirectional Convolutional LSTM (BiConvLSTM) architecture for extracting the long-term movement information present in the clips.

Although most of these techniques employ the raw RGB video stream as the input, we probed these architectures by also feeding them with the so-called *frame-difference* video stream, i.e., the difference in the adjacent frames. Frame differences serve as an efficient alternative to the computationally expensive optical flow. It is shown to be effective in several previous works [30,32,33] by promoting the model to encode the temporal changes between the adjacent frames, boosting the capture of the motion information.

### 4.2. Experimental Setting

We exploited three different, very general violence detection datasets to train the above methods: *Surveillance Camera Fight* [21], *Real-Life Violence Situations* [23], and *RWF-2000* [22], already mentioned in Section 2 and summarized in Table 1. *Surveillance Camera Fight* contains 300 videos, while both *Real-Life Violence Situations* and *RWF-2000* contain 2000 videos. All these datasets are perfectly balanced with respect to the number of violent and non-violent shots. The scenes captured in these datasets, recorded from fixed security cameras, collect very heterogeneous and everyday-life violent and non-violent actions. Therefore, they are the best candidate datasets available in the literature to train deep neural networks to recognize general violent actions. Other widely used datasets, such as *Hockey Fight* [16] or *Movies Fight* [16], do not contain enough diverse violence scenarios that can be transferable to public transport scenarios, and therefore we discarded them in our analysis.

Concerning the action recognition models, we replaced the final classification head with a binary classification layer, outputting the probability that the given video contains (or does not contain) violent actions. To obtain a fair comparison among all the considered methods, we employed their original implementations in PyTorch if present, and we re-implemented them otherwise. Moreover, when available, we started from the models pre-trained on *Kinetics-400*, the common dataset used for training general action recognition models.

Following previous works, we used *Accuracy* to measure the performance of the considered methods, defined as:(1)$$Accuracy=\frac{TP+TN}{TP+TN+FP+FN},$$
where TP, TN, FP, and FN are the true positives, true negatives, false positives, and false negatives, respectively. To have a more in-depth comparison between the obtained results, we also considered as metrics the *F1-score*, the *false alarm*, and the *missing alarm*, defined as follows:(2)$$F1=2\times \frac{Precision\times Recall}{Precision+Recall},$$
(3)$$FalseAlarm=\frac{FP}{TN+FP},$$
(4)$$MissingAlarm=\frac{FN}{TP+FN},$$
where Precision and Recall are defined as $\frac{TP}{TP+FP}$ and $\frac{TP}{TP+FN}$, respectively. Finally, to account also for the probabilities of the detections, we employed the *Area Under the Receiver Operating Characteristics (ROC AUC)*, computed as the area under the curve plotted with the true positive rate (TPR) against the false positive rate (FPR) at different threshold settings, where $TPR=Recall=\frac{TP}{TP+FN}$ and $FPR=\frac{FP}{TN+FP}$.

We employed the following evaluation protocol to have reliable statistics on the final metrics. For each of the three considered training datasets, we randomly varied the training and validation subsets five times, picking up the best model in terms of the accuracy and testing it over the full *Bus Violence* benchmark. Then, we reported the mean and the standard deviation of these five runs.

### 4.3. Results and Discussion

We report the results obtained by exploiting the three training general violence detection datasets in Table 3, Table 4 and Table 5. Considering the pre-training *Surveillance Camera Fight* dataset, the model which turns out to be the most performing is SlowFast, followed by the Video Swim Transformer. On the other hand, regarding the *Real-life Violence Situations* dataset in Table 4, the best model was the ResNet 3D network, followed by SlowFast. Finally, concerning the *RWF-2000* benchmark (Table 5), the more accurate models are the ResNet 2+1D, the SlowFast, and the Video Swim Transformer architectures. However, overall, all the considered models exhibit a moderate performance, indicating the difficulties in generalizing their abilities in classifying videos in the new challenging scenario represented by our *Bus Violence* dataset.

An important observation can be made concerning *false alarms* and *missing alarms*. Specifically, while all the considered methods generally obtained very good results regarding the first metric, they struggled with the latter. Because missing alarms are critical in this use-case scenario, because they reflect violent actions that happened but were not detected, this represents a major limitation for all the state-of-the-art violence detection systems. The main method responsible for this problem is to be sought in the high number of false negatives, which indeed also affects the Recall and, consequently, the F1-score, another evaluation metric that is particularly problematic for all the considered methods. In Figure 3, we report some samples of a true positive, true negative, false positive, and false negative. Another point worthy of note is that the majority of the most performing methods come from the human action recognition task. We deem that they are more robust to generalization to unseen scenarios because they are pre-trained using the *Kinetics-400* dataset, from which they learned more strong features able to help the network in classifying the videos also in this specific use case.

Finally, we report in Figure 4 the ROC curves concerning the three most performing models, i.e., SlowFast, ResNet 3D, and Video Swin Transformer, considering both the color and frame-difference inputs. Specifically, we plotted the curves for all three employed pre-training datasets. The dataset which provides the best generalization capabilities over our *Bus Violence* benchmark resulted in being the *Surveillance Camera Fight* dataset, followed by *RWF-2000*. However, as already highlighted, not one architecture shines when tested against our challenging scenario.

## 5. Conclusions and Future Directions

In this paper, we proposed and made freely available a novel dataset, called *Bus Violence*, which collects shots from surveillance cameras inside a moving bus, where some actors simulated both violent and non-violent actions. It is the first collection of videos describing violent scenes on public transport, characterized by peculiar challenges, such as different backgrounds due to the bus movement and illumination changes due to the varying positions of the vehicle. This dataset has been proposed as a benchmark for testing the current state-of-the-art violence detection and action detection networks in challenging public transport scenarios. This research is motivated by the fact that public transports are very exposed to many violent or criminal situations, and their automatic detection may be helpful to trigger an alarm to the local authorities promptly. However, it is known that state-of-the-art deep learning methods cannot generalize well to never seen scenarios due to the Domain Shift problem, and specific data are needed to train architectures to work correctly on the target scenarios.

In our work, we verified many state-of-the-art video-based architectures by training them on the largely used violence datasets (*Surveillance Camera Fight*, *Real-life Violence Situations*, and *RWF-2000*), and then testing them on the collected *Bus Violence* benchmark. The performed experiments showed that even very recent networks—such as Video Swin Transformers—could not generalize to an acceptable degree, probably due to the changing lighting and environmental conditions, as well as difficult camera angles and low-quality images. The CNN-based approaches seem to obtain the best results, still reaching an unsatisfactory level to make such systems reliable in real-world applications.

From our findings, we can conclude that the probed architectures cannot generalize to conceptually similar yet visually different scenarios. Therefore, we hope that the provided dataset will serve as a benchmark for training and/or evaluating novel architectures able to also generalize to these particular yet critical real-world situations. In this regard, we claim that domain-adaptation techniques are the key to obtaining features not biased to a specific target scenario [34,35]. Furthermore, we hope that the rising research in unsupervised and self-supervised video understanding [36,37] can be a good direction for acquiring high-level knowledge directly from pixels, without any manual or automatic labeling. This would pave the way toward plug-and-play smart cameras capable of learning about the specific scenario once deployed in the real world.

Finally, we also plan to use the acquired dataset for other relevant tasks on public transport, such as left-object detection and people counting, and to extend the collected videos to include other critical scenarios, such as unexpected emergencies—heart or panic attacks, that could be misclassified as some violent actions.

## Figures and Tables

**Figure 1 sensors-22-08345-f001:**
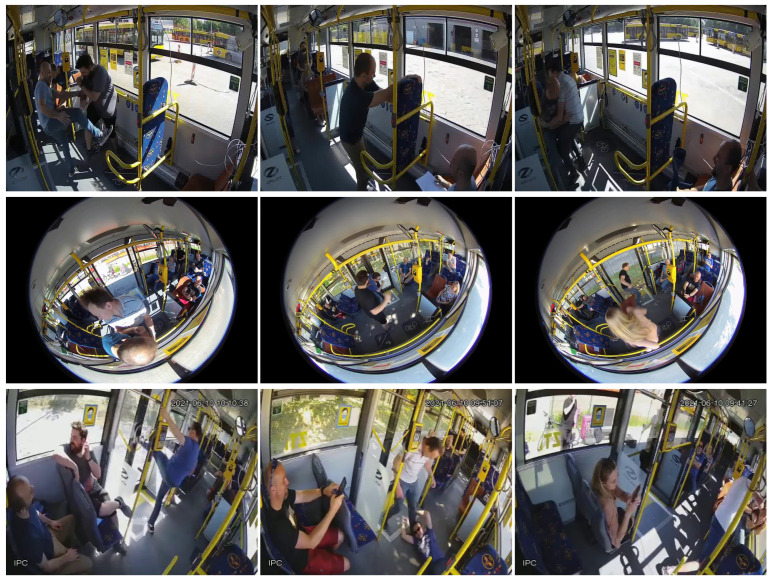
Samples of our *Bus Violence* benchmark belonging to the *violence* class, where the actors simulated violent actions, such as fighting, kicking, or stealing an object from another person. Each row corresponds to a different camera having a different perspective.

**Figure 2 sensors-22-08345-f002:**
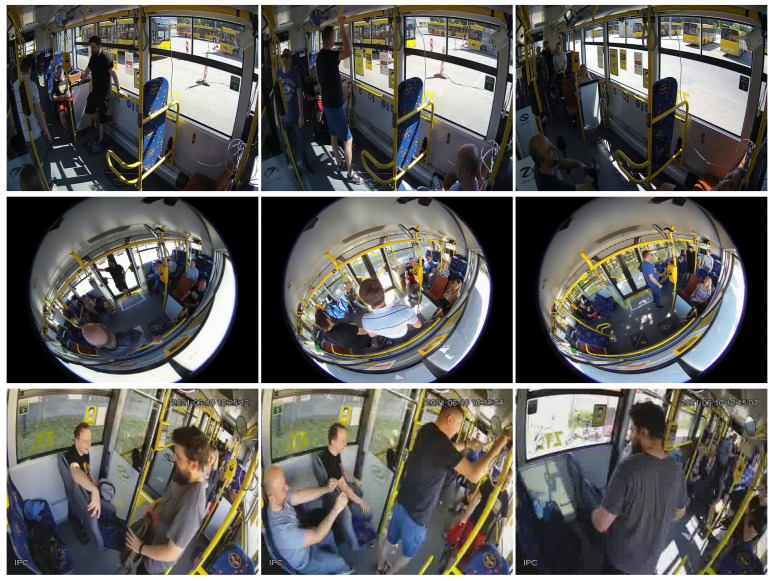
Samples of our *Bus Violence* benchmark belonging to the *non-violence* class, where the actors were just sitting, standing, or walking inside the bus. Each row corresponds to a different camera having a different perspective.

**Figure 3 sensors-22-08345-f003:**
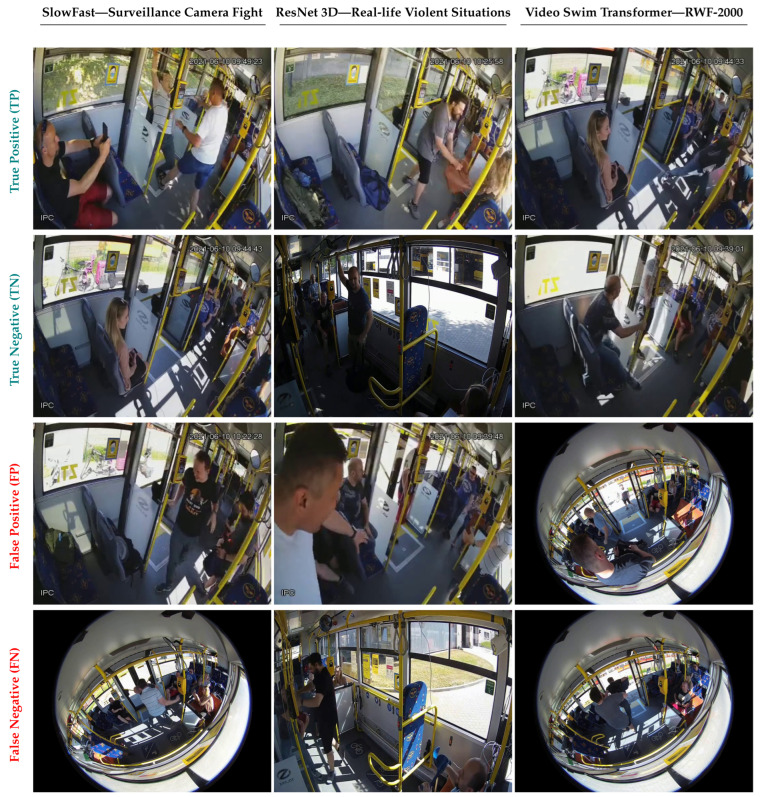
Some samples of predictions concerning the three most performing pairs models/pre-training datasets, i.e., SlowFast/Surveillance Camera Fight, ResNet 3D/Real-life Violent Situations, and Video Swim Transformer/RWF-2000 (one for each column). In the four rows, we report true positives, true negatives, false positives, and false negatives.

**Figure 4 sensors-22-08345-f004:**
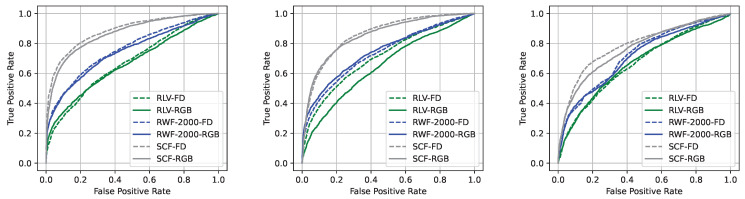
ROC curves concerning the three most performing pairs models/pre-training datasets, i.e., SlowFast/Surveillance Camera Fight SCF), ResNet 3D/Real-life Violent Situations (RLV), and Video Swim Transformer/RWF-2000, tested against our *Bus Violence* benchmark. We report the curves for both the color (RGB) and frame-difference (FD) inputs.

**Table 1 sensors-22-08345-t001:** Summary of the most popular existing datasets in the literature. We report the task for which they are used, together with the number of classes and videos that characterized them.

Name of Dataset	Task	Number of Classes	Number of Videos
Kinetics 400/600/700 [10,11,12]	Human Action Detection	400/600/700	650,000
HMDB51 [13]	Human Action Detection	51	7000
UCF-101 [14]	Human Action Detection	101	13,000
UCF-Crime [15]	Anomaly Detection	13	1900
NTU CCTV-Fights [18]	Violence Detection	2	1000
AIRTLab [19,20]	Violence Detection	2	350
Hockey and Movies Fight [16]	Violence Detection	2	1000
Violent-Flows [17]	Violence Detection	2	250
Surveillance Camera Fight [21]	Violence Detection	2	300
RWF-2000 [22]	Violence Detection	2	2000
Real-Life Violence Situations [23]	Violence Detection	2	2000

**Table 2 sensors-22-08345-t002:** The basic information concerning the *Bus Violence* benchmark, including the number of situations for violent and non-violent actions and the basic number of frames.

		# Videos with Resolution		
Class	# Situations	1280×960 px	960×540 px	352×288 px
Violence	700	212	222	266
Non-violence	700	240	210	250

**Table 3 sensors-22-08345-t003:** Cross-dataset evaluation (pre-training on *Surveillance Camera Fight* [21] dataset, test on our *Bus Violence* dataset).

		*Accuracy*↑	*F1*↑	*False Alarm*↓	*Miss Alarm*↓	*ROC AUC*↑
Model	Mode					
Hanson et al. [32] *	color-rgb	0.5383 ± 0.0236	0.1894 ± 0.1169	0.0362 ± 0.0303	0.8871 ± 0.0753	0.6813 ± 0.0274
	frame-diff	0.5175 ± 0.0166	0.1907 ± 0.1622	0.0975 ± 0.1329	0.8675 ± 0.1305	0.6105 ± 0.0668
Sudhakaran and Lanz [30]	color-rgb	0.5236 ± 0.0098	0.2729 ± 0.1887	0.1654 ± 0.1721	0.7875 ± 0.1835	0.5517 ± 0.0192
	frame-diff	0.5250 ± 0.0136	0.3495 ± 0.1773	0.2348 ± 0.1504	0.7152 ± 0.1749	0.5432 ± 0.0137
ResNet 2+1D [24] ^$^	color-rgb	0.6620 ± 0.0602	0.5063 ± 0.1355	0.0393 ± 0.0308	0.6368 ± 0.1396	0.7915 ± 0.0495
	frame-diff	0.6396 ± 0.0714	0.4488 ± 0.1831	0.0382 ± 0.0276	0.6825 ± 0.1695	0.8087 ± 0.0364
ResNet 3D [24] ^$^	color-rgb	0.6780 ± 0.0399	0.5417 ± 0.0938	0.0334 ± 0.0145	0.6106 ± 0.0937	0.8745 ± 0.0057
	frame-diff	0.6555 ± 0.0349	0.4929 ± 0.0788	0.0286 ± 0.0076	0.6604 ± 0.0759	0.8622 ± 0.0222
SlowFast [27] ^$^	color-rgb	0.7596 ± 0.0509	0.6955 ± 0.0999	0.0606 ± 0.0669	0.4203 ± 0.1674	0.8963 ± 0.0079
	frame-diff	0.7597 ± 0.0548	0.6896 ± 0.1083	0.0360 ± 0.0260	0.4446 ± 0.1328	0.8955 ± 0.0203
Video Swim Transformer [28] ^$^	color-rgb	0.6721 ± 0.0404	0.5596 ± 0.0824	0.0814 ± 0.0557	0.5743 ± 0.1030	0.7864 ± 0.0443
	frame-diff	0.6971 ± 0.0547	0.5972 ± 0.1373	0.0839 ± 0.0735	0.5218 ± 0.1706	0.8065 ± 0.0473

* Re-implemented in this work. ^$^ Pre-trained on *Kinetics-400*.

**Table 4 sensors-22-08345-t004:** Cross-dataset evaluation (pre-training on *Real-life Violence Situations* [23] dataset, test on our *Bus Violence* dataset).

		*Accuracy*↑	*F1*↑	*False Alarm*↓	*Miss Alarm*↓	*ROC AUC*↑
Model	Mode					
Hanson et al. [32] *	color-rgb	0.5846 ± 0.0212	0.4976 ± 0.0905	0.2597 ± 0.1220	0.5711 ± 0.1389	0.6150 ± 0.0068
	frame-diff	0.5787 ± 0.0268	0.3786 ± 0.0957	0.1079 ± 0.0539	0.7346 ± 0.1003	0.6385 ± 0.0366
Sudhakaran and Lanz [30]	color-rgb	0.5195 ± 0.0021	0.4482 ± 0.0261	0.3529 ± 0.0405	0.6082 ± 0.0421	0.5533 ± 0.0245
	frame-diff	0.5420 ± 0.0208	0.5166 ± 0.0996	0.4337 ± 0.1772	0.4823 ± 0.1823	0.5608 ± 0.0210
ResNet 2+1D [24] ^$^	color-rgb	0.5938 ± 0.0913	0.3660 ± 0.2960	0.1081 ± 0.1079	0.7043 ± 0.2881	0.7108 ± 0.0665
	frame-diff	0.5576 ± 0.0222	0.2650 ± 0.1142	0.0540 ± 0.0626	0.8309 ± 0.0978	0.6723 ± 0.0700
ResNet 3D [24] ^$^	color-rgb	0.6021 ± 0.0399	0.4739 ± 0.1493	0.1920 ± 0.2021	0.6037 ± 0.1971	0.6728 ± 0.0254
	frame-diff	0.6521 ± 0.0265	0.6333 ± 0.0640	0.3186 ± 0.1839	0.3771 ± 0.1769	0.7334 ± 0.0468
SlowFast [27] ^$^	color-rgb	0.5976 ± 0.0497	0.3495 ± 0.1552	0.0383 ± 0.0423	0.7666 ± 0.1402	0.6794 ± 0.0215
	frame-diff	0.5704 ± 0.0143	0.2833 ± 0.0698	0.0318 ± 0.0296	0.8273 ± 0.0531	0.6616 ± 0.0287
Video Swim Transformer [28] ^$^	color-rgb	0.5769 ± 0.0421	0.3278 ± 0.1572	0.0714 ± 0.0677	0.7749 ± 0.1408	0.6875 ± 0.0456
	frame-diff	0.6130 ± 0.0498	0.4992 ± 0.1863	0.2077 ± 0.1585	0.5663 ± 0.2107	0.6771 ± 0.0590

* Re-implemented in this work. ^$^ Pre-trained on *Kinetics-400*.

**Table 5 sensors-22-08345-t005:** Cross-dataset evaluation (pre-training on *RWF-2000* [22] dataset, test on our *Bus Violence* dataset).

		*Accuracy*↑	*F1*↑	*False Alarm*↓	*Miss Alarm*↓	*ROC AUC*↑
Model	Mode					
Hanson et al. [32] *	color-rgb	0.5120 ± 0.0078	0.0690 ± 0.0357	0.0126 ± 0.0058	0.9634 ± 0.0200	0.6692 ± 0.0506
	frame-diff	0.5041 ± 0.0050	0.0272 ± 0.0200	0.0029 ± 0.0023	0.9889 ± 0.0109	0.6044 ± 0.0212
Sudhakaran and Lanz [30]	color-rgb	0.5109 ± 0.0100	0.0868 ± 0.0859	0.0280 ± 0.0329	0.9503 ± 0.0524	0.5230 ± 0.0257
	frame-diff	0.5024 ± 0.0019	0.0261 ± 0.0153	0.0086 ± 0.0077	0.9866 ± 0.0081	0.5287 ± 0.0183
ResNet 2+1D [24] ^$^	color-rgb	0.5477 ± 0.0232	0.1788 ± 0.0766	0.0049 ± 0.0030	0.8997 ± 0.0472	0.7085 ± 0.0618
	frame-diff	0.5806 ± 0.0192	0.2868 ± 0.0579	0.0089 ± 0.0037	0.8300 ± 0.0402	0.7607 ± 0.0440
ResNet 3D [24] ^$^	color-rgb	0.5540 ± 0.0176	0.1997 ± 0.0571	0.0043 ± 0.0010	0.8877 ± 0.0353	0.7645 ± 0.0224
	frame-diff	0.5383 ± 0.0201	0.1456 ± 0.0681	0.0034 ± 0.0013	0.9200 ± 0.0405	0.7515 ± 0.0264
SlowFast [27] ^$^	color-rgb	0.5856 ± 0.0275	0.2936 ± 0.0805	0.0037 ± 0.0019	0.8251 ± 0.0563	0.7849 ± 0.0569
	frame-diff	0.5596 ± 0.0259	0.2141 ± 0.0779	0.0029 ± 0.0017	0.8780 ± 0.0516	0.7922 ± 0.0289
Video Swim Transformer [28] ^$^	color-rgb	0.5496 ± 0.0157	0.2024 ± 0.0563	0.0161 ± 0.0088	0.8846 ± 0.0362	0.7313 ± 0.0642
	frame-diff	0.5618 ± 0.0347	0.2329 ± 0.1142	0.0143 ± 0.0098	0.8621 ± 0.0781	0.7441 ± 0.0636

* Re-implemented in this work. ^$^ Pre-trained on *Kinetics-400*.

## Data Availability

The dataset is freely available at https://zenodo.org/record/7044203#.Yxm7hmxBxhE (8 September 2022).

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
