# Peer review of "Bus Violence: An Open Benchmark for Video Violence Detection on Public Transport"

_sensors, 2022, doi:10.3390/s22218345_

Round 1
Reviewer 1 Report
The paper has a practical character, it aims at raising the quality of life of people in public transportation, namely busses. Results of this benchmark are surely worth presenting and publishing in the journal. It is important that the Authors have previously released their own dataset [9], which is available for other researchers around the world.
The manuscript is written in proper English, it is informative and pleasant to read.
Suggestions and comments:
Check the Authors and Affiliation section – there is not e-mail for co-authors from Institution 2, 3, and 4. Moreover, in case of no. 4 you need to type in the post code, street, etc.
It would be better to highlight the novelty of your paper using bullet points.
Authors could provide additional information on their dataset, e.g., file format, resolution, bitrate, bit depth, length (seconds, minutes), etc. How many of them were available at 960x540 and 1280x960? Were they all MP4 (H264)? Moreover, were their obtained using the equipment installed in busses themselves (by the public transportation agency)? Or did they come from another government or NGO institution?
Reformat the main body of the manuscript in order to avoid blank spaces (e.g., see page 9).
After reading the full paper, I would encourage authors to extend the Conclusions and Future Works part. Do mention about future study directions. Provide additional feedback and source of inspiration for other researchers and potential readers. Highlight, once again, why is you dataset unique. What makes it different from the others? Why should scholars choose this one instead of others.
The paper is quite short, therefore I would advise the Authors to insert more figures/screenshots summing up their dataset. Do you have content from different parts of the day (daytime, nighttime)? With people wearing different clothes (colder, warmer temperature, etc.). Scenes involving a single, a couple, multiple, and dozens (multiple) of individuals?
Consider extending the number of cited references. Currently, approx. 30 is a quite fair number. However, taking into account the topic of this paper, it would be advisable to look for additional journal papers and conference proceedings.
At the end, how did you classify your content? Was it performed by one or many human individuals? Or maybe the labelling was automated in some part? If yes, what kind of software, toolboxes, libraries, etc., did you utilize?
This is a good paper, but it deserves to be a very good one. Therefore, some modifications and extensions are suggested.
Reviewer 2 Report
Violence detection is a significant and meaningful issue in research area of vadio image process. This paper proposes a novel benchmark for video violence detection in public transport, which will be very helpful for video surveillance in this specific scenario. There still exists some weaknesses as follows need to be enhanced.
(1) This research work supposes that surveillance videos in public transport only cover two types, i.e. violence and non-violence. The established video dataset are classified as these two types as well. However, in some abnormal scenarios, it is be quite diffcult to decisid whether there exists violent action in it, indeed. For example, an unexpected emergence treatment for unsettled heart attack could be misclassified as some violent action. Therefore, these abnormal scenarios should be contained in your video dataset as well, theoretically. If seldom of these are contained, it is suggested that this problem should be considered in future research in conclusions.
(2) All numbers of means and the standard deviations in Table 3-5 are advised to be accurate to four decimal places, as the convention in ML.
Round 2
Reviewer 1 Report
This is a very good paper, the topic is interesting and has a practical character. Personally, I do favor joint actions among many domestic as well as different international institutions. Please do continue your studies and keep up with presenting and publishing your results in conference proceedings as well as journals.